# The Potential Use of Fungal Co-Culture Strategy for Discovery of New Secondary Metabolites

**DOI:** 10.3390/microorganisms11020464

**Published:** 2023-02-12

**Authors:** Shuang Xu, Mengshi Li, Zhe Hu, Yilan Shao, Jialiang Ying, Huawei Zhang

**Affiliations:** School of Pharmaceutical Sciences, Zhejiang University of Technology, Hangzhou 310014, China

**Keywords:** fungus, co-culture, secondary metabolite, biosynthetic gene cluster, interaction mechanism

## Abstract

Fungi are an important and prolific source of secondary metabolites (SMs) with diverse chemical structures and a wide array of biological properties. In the past two decades, however, the number of new fungal SMs by traditional monoculture method had been greatly decreasing. Fortunately, a growing number of studies have shown that co-culture strategy is an effective approach to awakening silent SM biosynthetic gene clusters (BGCs) in fungal strains to produce cryptic SMs. To enrich our knowledge of this approach and better exploit fungal biosynthetic potential for new drug discovery, this review comprehensively summarizes all fungal co-culture methods and their derived new SMs as well as bioactivities on the basis of an extensive literature search and data analysis. Future perspective on fungal co-culture study, as well as its interaction mechanism, is supplied.

## 1. Introduction

Microbial secondary metabolites (SMs) are an important source for developing drugs and biopesticides, and approximately 61.5% of microbial SMs are derived from fungal strains [1]. Fungal SMs structurally consist of polyketides, terpenes, and alkaloids with a wide array of biological properties, such as antibacterial, antifungal, or antitumour activity [2,3]. These substances have played, and continue to play, a crucial role in the discovery of new drugs, such as penicillin, lovastatin, cyclosporine, and so on [3,4,5,6]. With the falling costs of sequencing technology in the post-genome era, whole-genome sequencing and functional genome mining techniques have been widely employed, and have unveiled thousands of microbial biosynthetic gene clusters (BGCs) responsible for producing cryptic SMs [7,8]. Since fungal strains are favorable to traditional culture conditions, most of their BGCs are unawaken or expressed in very low levels [9,10,11]. Therefore, the number of new fungal SMs by traditional axenic culture method has been greatly decreasing in the past two decades despite great progress made in chemical analysis and separation techniques. Fortunately, several approaches have been successfully developed and implemented for interrogating these silent BGCs and enhancing chemical diversity of fungal SM, such as one strain many compounds (OSMAC) strategy [12,13,14], heterologous expression technique [15,16,17,18,19], promoter engineering approach [20,21,22,23], and other genetic engineering methods [24,25,26,27,28]. A growing number of studies suggest that microbial co-culture has a greater effect not only on microbe growth but also on metabolism than axenic culture [11,29]. As one of the most commonly used OSMAC methods, co-culture is a simple and highly efficient tool, used to activate silent BGCs for production of novel SMs (Figure 1).

By simulating naturally occurring conditions, microbial co-culture strategy can improve antibiotic activity in crude extracts, increase the yield of known SMs, produce analogues of known metabolites, and induce previously unexpressed bioactive ingredient pathways [11,30,31,32]. Fungal co-culture strategy is an effective approach to awakening silent BGCs in fungal strains to produce cryptic SMs and this strategy usually consists of three approaches, including fungal–fungal, fungal–bacterial, and fungal–host co-cultures. In order to enrich our knowledge of this strategy and to use it to obtain new SMs, this review comprehensively summarized all fungal co-culture methods and their derived new SMs, as well as biological activities, for the first time. By extensive literature search using DNP (Dictionary of Natural Products) database and SciFinder tool, as many as **158** (**1**–**158**) new SMs were isolated and structurally characterized using fungal co-culture strategy until now. Based on fungal co-culture type, these new substances are respectively introduced herein and their detailed information is supplied in Appendix A.

## 2. Fungal–Fungal Co-Culture

Fungal–fungal co-culture is the major source of new SMs (**1**–**109**) and consists of two types including liquid state fermentation (LSF) and solid state fermentation (SSF). Potato dextrose broth (PDB) and rice are the most common co-culture media for fungal LSF and SSF, respectively.

### 2.1. Liquid State Fermentation (LSF)

Although the growth interference between two co-cultured fungal strains is very complex, it is well recognized that LSF can facilitate metabolite transportation and exchange, and trigger new SMs biosynthesis [33]. By the end of 2022, a total of 75 new SMs (**1**–**75**) had been obtained from LSF co-culture, such as polyketides, macrolides, terpenes, etc.

#### 2.1.1. Potato Dextrose Broth (PDB)

PDB is the most widespread used medium for growing fungi under aerobic condition. Among these LSF-derived new SMs, 54 compounds (**1**–**54**, Figure 2) were discovered in PDB using fungal–fungal co-culture. Chemical investigation of the co-culture broth of *Nigrospora oryzae* and *Beauveria bassiana* led to isolation of five new azaphilones (**1**–**5**), in which compound **2** had an unprecedented skeleton with a bicyclic oxygen bridge and compounds **4** and **5** showed significant nitric oxide (NO) inhibitory activity at 50 µM with inhibition rates of 37% and 39%, respectively [34].

Two new aryl esters (**6** and **7**) and six new protoilludane-type sesquiterpenes (**8**–**13**) were produced by *Armillaria* sp. when co-cultured with an endophytic fungus *Epicoccum* sp., and compound **13** showed moderate in vitro cytotoxic activities toward human cancer cell lines (HL-60, A549, MCF-7, SMMC-7721, and SW480) with IC_50_ values ranging from 15.80 to 23.03 μM and weak inhibitory activity against acetylcholinesterase (AChE) [35]. The co-culture of *N. oryzae* and *Irpex lacteus* from seeds of *Dendrobium officinale* resulted in production of four new SMs (**14**–**17**) belonging to two backbones of pulvilloric acid-type azaphilone as well as a tremulane sesquiterpene (**18**) equipped with strong anti-AChE activity at 50 μM [36]. Three new zinniol analogues, pleoniols A–C (**19**–**21**), were detected in the co-culture extract of two endophytic strains *Pleosporales* sp. F46 and *Acremonium pilosum* F47 [37]. The co-culture of two symbiotic fungi *Phoma* sp. YUD17001 and *Armillaria* sp. derived from *Gastrodia elata* afforded four polyketones (**22**–**25**) and a new nitrogenous compound (**26**) [38]. Phomretones A–F (**27**–**32**) were new bicyclic polyketides purified from the co-culture of *Armillaria* sp. and the endophytic strain *Phoma* sp. YUD17001 [39]. The co-culture of *Penicillium fuscum* and *P. camembertii*/*clavigerum* afforded the production of eight new 16-membered-ring macrolides (**33**–**40**), of which compound **33** exhibited the most potent antimicrobial activity against MRSA strains as well as *Bacillus anthracis*, *Streptococcus pyogenes*, *Candida albicans*, and *C. glabrata*. The mechanism of action (MoA) study indicated that **33** did not inhibit bacterial protein synthesis nor target their ribosomes, suggesting a novel mode of action for its antibiotic activity [40]. A novel alkaloid named harziaphilic acid (**41**) was produced in the co-culture of two plant beneficial fungi *Trichoderma harzianum* M10 and *Talaromyces pinophilus* F36CF and displayed selectively inhibitory effect on the proliferation of cancer cells [41]. Co-cultivation of *Aspergillus nidulans* with *Epicoccum dendrobii* produced eight new SMs (**42**–**49**) as well as six known compounds. The mechanisms that trigger metabolic changes during fungal–funal interactions was determined that VeA1 regulation requires the transcription factor SclB and velvet complex members LaeA and VelB to generate aspernidines as representative formation of SM in *A. nidulans* [1]. When co-cultured with *Botrytis cinerea*, the biocontrol fungus *Purpureocillium lilacinum* was shown to manufacture a new unusual linear polypeptide leucinostatin (**50**) in PDB detected by matrix-assisted laser desorption ionization-time of flight mass spectrometry imaging mass spectrometry (MALDI-TOF-IMS) [42]. The co-culture of *Phellinus orientoasiaticus* and *Xylodon flaviporus* yielded three new sesquiterpenes (**51**–**53**) [43]. One new 10-membered lactone (**54**) was separated from the co-culture of *Nigrospora* sp. and *Stagonosporopsis* sp. and exhibited antifungal activities against *P. janthinellum*, *Aspergillus fumigatus*, *Phomopsis* sp. and *Alternaria* sp. [44].

#### 2.1.2. Other Liquid Media

Several other liquid media were uncommonly used in fungal–fungal co-culture system and resulted in production of 21 new SMs (**55**–**75**, Figure 3). Pleurotusins A (**55**) and B (**56**) along with five known terpenoids were produced by two edible fungi *Pleurotus ostreatus* SY10 and *P. eryngii* SY302 when co-cultivated in liquid medium consisting of glucose 10 g/L, KH_2_PO_4_ 1 g/L, MgSO_4_ 0.5 g/L, peptone 2 g/L, and 1 L sterilized water [45,46]. A new *N*-methoxypyridone analog (**57**) was not produced by the monoculture of the two endophytic strains *Camporesia sambuci* FT1061 and *Epicoccum sorghinum* FT1062, but was synthesized in the co-culture, which consisted of mannitol 20 g/L, sucrose 10 g/L, monosodium glutamate 5 g/L, KH_2_PO_4_ 0.5 g/L, MgSO_4_·7 H_2_O 0.3 g/L, yeast extract 3 g/L, corn steep liquor 2 mL/L, and 1 L distilled water [47]. Owing to the appearance of pigments in adversarial zones between *Penicillium pinophilum* FKI-5653 and *Trichoderma harzianum* FKI-5655 on PDA plate, chemical study of their co-culture resulted in the isolation of a novel diphenyl ether (**58**) from glucose–peptone broth [48]. A new chlorinated bianthrone (**59**) was isolated from the co-culture of two different developmental stages of a marine alga-derived *Aspergillus alliaceus* strain in malt liquid medium and showed weak cytotoxic activity against the HCT-116 colon cancer and SK-Mel-5 melanoma cell lines [49]. This is the first example of self-induced metabolomic changes resulting in the production of allianthrones. Fungal strains *Chaunopycnis* sp. CMB-MF028 and *T. hamatum* CMB-MF030 were co-isolated from the inner tissue of an intertidal rock platform mollusc; co-cultivation of these fungi in ISP2 broth resulted in transcriptional activation of a new 2-alkenyl-tetrahydropyran (**60**) [50].

Sclerotiorumins A–C (**61**–**63**) together with one pyrrole derivative 1-(4-benzyl-1H-pyrrol-3-yl) ethanone (**64**) and two metallo-organic complexes (**65** and **66**) were synthesized by two marine-derived fungi *A. sclerotiorum* and *P. citrinum* in the co-culture liquid medium which consisted of glucose, soluble starch, MgSO_4_, KH_2_PO_4_, peptone, and sea salt [51]. When strains *Trametes versicolor* and *Ganoderma applanatum* co-cultured in a medium containing glucose (10 g/L), KH_2_PO_4_ (1 g/L), MgSO_4_ (0.5 g/L), and peptone (2 g/L)*,* two novel formamide derivatives (**67** and **68**) were highly synthesized and compound **68** displayed the potential to enhance the cell viability of a human immortalized bronchial epithelial cell line [52]. A novel alkaloid named as aspergicin (**69**) was isolated from the co-cultured mycelia of two marine-derived mangrove epiphytic *Aspergillus* sp. in GYP medium [53]. Two new antibacterial 1-isoquinolone analogs (**70** and **71**) were produced by two mangrove endophytic fungi (strain Nos. 1924 and 3893) in a co-culture system which consisted of glucose 10 g/L, peptone 2 g/L, yeast extracts 1 g/L, crude marine salt 3.5 g/L, and water 1 L [54,55]. A new diimide derivative (**72**) and three new cyclic peptides (**73**–**75**) were isolated and characterized from the co-culture of two mangrove fungi *Phomopsis* sp. K38 and *Alternaria* sp. E33 in liquid medium (glucose 10 g/L, peptone 2 g/L, yeast extract 1 g/L, NaCl 30 g/L), and compound **72** had weak cytotoxic activity against Hep-2 and HepG2 cells and compounds **73**–**75** exhibited moderate to high antifungal activities as compared with the positive control (Ketoconazole) [55,56,57].

### 2.2. Solid State Fermentation (SSF)

Metabolite exchange between co-cultured strains is limited in SSF and could reduce direct growth competition or interference between co-cultured members and facilitate co-cultured SMs biosynthesis [33]. By the end of 2022, 34 new SMs (**76**–**109**, Figure 4, Figure 5 and Figure 6) were obtained from fungal–fungal co-culture using SSF method. Rice and potato dextrose agar (PDA) are the most commonly used SSF media for fungal–fungal co-culture.

#### 2.2.1. Rice Solid Medium

Chlorotetralone (**76**) was a new antifungal aromatic polyketide produced by the endophytic *N. oryzae* co-cultured with *B. bassiana* on rice solid medium [58]. Six new isoprenylated chromanes (**77**–**80**) including two new isoprenylated phenol glucosides (**81**–**82**) were obtained from the co-cultured rice medium of *Pestalotiopsis* sp. and *P. bialowiezense* [59,60], and compound **77** showed significant *β*-glucuronidase inhibitory potency [32]. Two new citrinin analogs (**83** and **84**) were detected in the co-culture of two marine algal-derived endophytic strains *A. sydowii* EN-534 and *P. citrinum* EN-535 [61]. Chermebilaene A (**85**) together with a new orthoester meroterpenoid (**86**) was the first natural sesquiterpene hybridized with octadecadienoic acid from the co-culture of two marine strains *P. bilaiae* MA-267 and *P. chermesinum* EN-480, and showed potent inhibitory activity against pathogenic fungi *Ceratobasidium cornigerum* and *Edwardsiella tarda* [62]. A new antimicrobial terrein derivative, namely asperterrein (**87**), was metabolized by a marine red alga-derived endophytic fungus *A. terreus* EN-539 co-cultured with symbiotic fungus *Paecilomyces lilacinus* EN-531, which was isolated from the inner tissues of the marine red alga *Laurencia okamurai* [63]. Five new prenylated indole alkaloids (**88**–**92**) were isolated from a co-culture of marine-derived fungi *A. sulphureus* KMM 4640 with *Isaria felina* KMM 4639 on rice solid medium, and compound **89** was able to inhibit the colony formation of human prostate cancer cells 22Rv1 at non-cytotoxic concentration of 10 μM [64]. Chemical investigation of a co-culture of the marine-derived fungi *I. felina* KMM 4639 and *A. carneus* KMM 4638 on rice led to the discovery of three new drimane-type sesquiterpenes (**93**–**95**) [65].

#### 2.2.2. Potato Dextrose Agar (PDA) 

Until now, only three chemical studies using PDA medium for fungal–fungal co-culture have been reported. Large-scale co-cultivation of *Plenodomus influorescens* and *Pyrenochaeta nobilis* on PDA resulted in isolation of a new azaphilone (**96**) and a new macrolides (**97**) [66]. *Trichophyton rubrum* and *Bionectria ochroleuca* were derived from different environments and were shown to generate a long-distance interaction zone, which led to the discovery of a substituted trimer of 3,5-dimethylorsellinic acid (**98**) [67]. Its nonsulfated form and three other known analogs were found in the monoculture of *B. ochroleuca*. Co-cultivation of *Cosmospora* sp. and *Magnaporthe oryzae* resulted in the production of two new dihydro-isocoumarins, soudanones H-I (**99**–**100**) on PDA medium [68].

#### 2.2.3. Other Solid Media

Several other solid media have been used for fungal–fungal co-culture followed by chemical investigation. Whelone (**101**) was a new polyketide isolated and identified from the co-culture of *A. fischeri* NRRL 181 and *T. labelliformis* G536 on oatmeal [69]. Four new alkyl aromatics, penixylarins A−D (**102**–**105**), were isolated from a co-culture of an Antarctic deep-sea-derived fungus *Penicillium crustosum* PRB-2 and the mangrove-derived fungus *Xylaria* sp. HDN13-249 on solid medium mainly consisting of soluble starch, yeast extract, sucrose, maltose, bean flour, peptone, and agar [70]. Biological tests indicated that compound **104** exhibited potential antituberculosis effects. Citrifelins A (**106**) and B (**107**) possessing a unique tetracyclic framework were separated from the co-culture of *Penicillium citrinum* and *Beauveria felina* in wheat bran broth medium (100 mL of naturally sourced and filtered seawater, 100 g of wheat bran, and 0.6 g of dried potato powder) and showed inhibitory activities against several human and aquatic pathogens [71]. Co-cultivation of strains *T. pinophilus* 17F4103 and *Paraphaeosphaeria* sp. 17F4110 on a malt extract agar medium led to production of a new γ-pyridone (**108**) and enhancement of the production of penicillone C and D [72]. A new isoindolinone alkaloid, irpexine (**109**), was detected in the co-culture of *I. lacteus* with *Phaeosphaeria oryzae*, which consisted of malt extract 20 g/L, peptone 5 g/L, agar 15 g/L, and 1 L deionized water. In addition, a known green pigment hypoxyxylerone was detected in the axenic culture of *I. lacteus*, but its production was markedly enhanced by the co-culture [73].

## 3. Fungal–Bacterial Co-Culture

In a fungal–bacterial co-culture system, fungus is commonly used as host strain and bacterium is the guest. Owing to their notable difference in culture conditions, appropriate adjustment was made, such as inoculation amount and order. The discovery of penicillin is probably one of the most important experiments in fungal–bacterial co-culture [74,75]. Until the end of 2022, 49 new SMs (**110**–**158**) derived from fungal–bacterial co-culture have been reported and were also divided into two groups, LSF and SSF. It is interesting that ISP2 broth and rice are respectively their most used media for fungal–bacterial co-culture.

### 3.1. Liquid State Fermentation

As many as 23 new SMs (**110**–**131**, Figure 7) have been isolated and characterized from fungal–bacterial co-culture using LSF. Blennolide K (**110**) was a new cytotoxic blennolide obtained from a co-culture of *Setophoma terrestris* and *Bacillum amyloliquifaciens* in PDB medium [76]. Two new sesquiterpenes (**111** and **112**) and two new de-O-methyllasiodiplodins (**113** and **114**) were detected in the co-cultured ISP2 broth of a mangrove endophytic *Trichoderma* sp. 307 and an aquatic pathogenic *Acinetobacter johnsonii* B2, and compounds **112** and **114** exhibited potent α-glucosidase inhibitory activity with IC50 values of 25.8 and 54.6 µM, respectively, which were more potent than the positive control (acarbose, IC50 = 703.8 µM) [77]. Libertellenones A–D (**115**–**118**) were new pimarane diterpenes with cytotoxicity isolated from the co-cultured YPM (yeast extract, peptone and mannitol) medium of an ascidian-derived strain *Libertella* sp. CNL-523 and a α-proteobacterium strain CNJ-328 [78].

The co-culture of *A. fumigatus* with *Streptomyces peucetius* in ISP2 broth led to the induction of production of two new amides (**119** and **120**), of which the later had significant activity against several NCI-60 cell lines [79]. The addition of bacterial strain *S. bullii* to an established culture of *A. fumigatus* MBC-F1-10 in ISP2 broth resulted in the biosynthesis of a new spiro-lactam (**121**), which was considered to be of fungal origin due to its chemical nature [80]. Two new diketopiperazine disulfide, glionitrins A (**122**) and B (**123**), were produced using a co-culture of strains *A. fumigatus* KMC-901 and *Sphingomonas* sp. KMK-001 in Czapek-Dox broth, and compound **122** showed potent submicromolar cytotoxic activity against human cancer cell lines (HCT-116, A549, AGS, DU145) and displayed significant antibiotic activity against a series of microbes including MRSA [81,82]. The co-culture of durum wheat plant roots-associated bacterium *Pantoea aggolomerans* and date palm leaves-derived fungus *Penicillium citrinum* in ISP2 broth led to the synthesis of two new pulicatin derivatives (**124** and **125**), which proved to exhibit potential antifungal effects [83]. Three new decalin-type tetramic acid analogs (**126**–**128**) were separated from the co-cultivated ISP2 broth of the fungus *Fusarium pallidoroseum* and the bacterium *Saccharopolyspora erythraea* [84]. The co-culture of the marine fungus *Emericella* sp. with the marine actinomycete *Salinispora arenicola* in YPM medium led to the production of two new antimicrobial cyclic depsipeptides (**129** and **130**) [85]. A fungal–bacterial community composed of *Cladosporium* sp. WUH1 (host strain) and *B. subtilis* CMCC(B) 63501 (guest strain) in PDA-LB (Luria-Bertani) liquid medium was found to produce a novel diphenyl ether (**131**) with polyhydroxy side chain [86], and its production may be related to the fact that dormant or inert biosynthetic pathways can enter through multiple chemical interactions.

### 3.2. Solid State Fermentation

As shown in Figure 8, all fungal–bacterial co-culture-derived SMs (**132**–**151**) using SSF are PKS and PKS-NRPS products. Co-cultivation of the endophytic fungus *F. tricinctum* and strain *S. lividans* TK24 on solid rice medium resulted in the accumulation of four new dimeric naphthoquinones (**132**–**135**) and a new lateropyrone (**136**), which were not detected in axenic fungal controls [87]. Two new dihydronaphthalenone diastereomers (**137** and **138**) were purified from the co-culture of the endophytic fungus *A. versicolor* KU258497 with *B. subtilis* 168 trpC2 on rice medium. Compound **138** showed moderate cytotoxic activity against the mouse lymphoma cell line L5178Y with an IC_50_ value of 22.8 μM [88]. Co-culture of *A. sydowii* and *B. subtilis* on PDA led to the production of three new amides (**139**–**141**) and a novel polyene (**142**) [89]. A new isoprenylated benzophenone named pestalone (**143**) was produced by a marine brown alga-derived strain *Pestalotia* sp. CNL-365 when co-cultured with an unidentified marine bacterium in YPG (yeast extract, peptone, glucose) medium, and exhibited moderate in vitro cytotoxicity and potent antibiotic activity against MRSA and vancomycin-resistant *Enterococcus faecium* (VREF) [90]. Three new SMs (**144**–**146**) were characterized from the co-culture of a fungal endophyte *F. tricinctum* and a bacterial strain *B. subtilis* 168 trpC 2 on rice, but were not detected in discrete fungal and bacterial controls [91]. Co-culture of strain *Actinomycete* sp. WAC 2288 with *Cryptococcus neoformans* on Bennett’s agar led to the synthesis of a large macrolactone (**147**), which displayed preferential killing activity against *C. neoformans* targeted its cellular membranes [92]. Four new SMs including a cyclic pentapeptide (**148**), one aflaquinolone (**149**) and two anthraquinones (**150** and **151**) were obtained from the co-culture of a sponge-associated fungus *A. versicolor* and *B. subtilis* on rice medium [93].

## 4. Fungal–Host Co-Culture

Endophytic microorganisms are ubiquitous and have been found in all species of plants studied to date. However, most of the endophyte-host relationships are not well understood. Until now, only a few chemical studies have been carried out on fungal–host co-culture and have afforded seven new SMs (**152**–**158**, Figure 9), including a new antifungal 2,4-cyclopentadiene-1-one (**152**) from the co-culture of endophyte−host (*N. oryzae*, *I. lacteus*, and the host plant *D. officinale*) in PDB and six new anti-feedant polyketides (**153**–**158**) from the co-culture of *P. verruculosa* and *D. officinale* in PDB [94,95]. The interaction between host plant *D. officinale* and *P. verruculosa* was shown to have an important induction on the anti-phytopathogenic metabolite productions [96].

## 5. Conclusions and Perspectives

In summary, all new SMs (**1**–**158**) derived from fungal co-culture method were comprehensively summarized. These SMs displayed a wide variety of biological activities with medicinal and therapeutic potential. Co-culture for the biosynthesis of new SMs has proven to be an effective method with outstanding results. However, analyses of typical fungal genomes have suggested that the biosythetic potential is vast and the number of fungal SMs is far more than the existing figure. More efforts should be made on chemical investigation using fungal co-culture strategy.

In addition to inducement of biosynthesis of new SMs, fungal co-culture has important roles in increasing activity of valuable enzymes (such as laccase, pectinase, etc. [97,98,99,100,101]) and improving yield of important products (such as ursolic acid, oleanolic acid, betulinic acid, etc. [91,102,103,104,105,106,107,108,109,110,111,112]). It offers the unique advantages of low cost and simple operation, and does not require expensive chemicals and complex gene-level manipulation. However, there are still some key challenges to this promising research approach that need to be addressed. For instance, the molecular mechanisms that trigger metabolic changes in fungal co-culture interactions have attracted the interest of researchers. Several studies have confirmed that direct physical contact in fungal co-culture is required to elicit the specific response [113,114,115]. These mechanisms provide the rationale for a wide range of co-culture-based natural product discovery studies. However, for many reported fungal co-cultures, the exact mechanism of SMs biosynthesis is largely unknown. The growth interference between two co-cultured fungal strains is very complex, co-culture still suffers from difficulties for large scale production and structural identification of these SMs. Successful co-culture often requires suitable paired strains, and culture conditions should allow for the growth needs of co-culture members. The development of multi-omics technology and gene engineering will definitely assist in elucidating these interaction mechanisms and biosynthetic pathways [33,116,117]. In the future, this strategy will continue to play an important role in novel SM discovery and drug development.

## Figures and Tables

**Figure 1 microorganisms-11-00464-f001:**
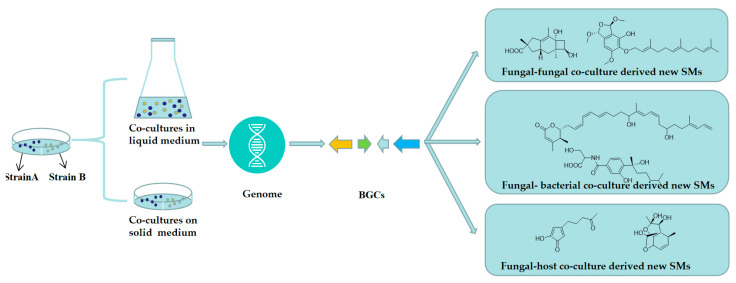
Flowchart for chemical study using fungal co-culture strategy.

**Figure 2 microorganisms-11-00464-f002:**
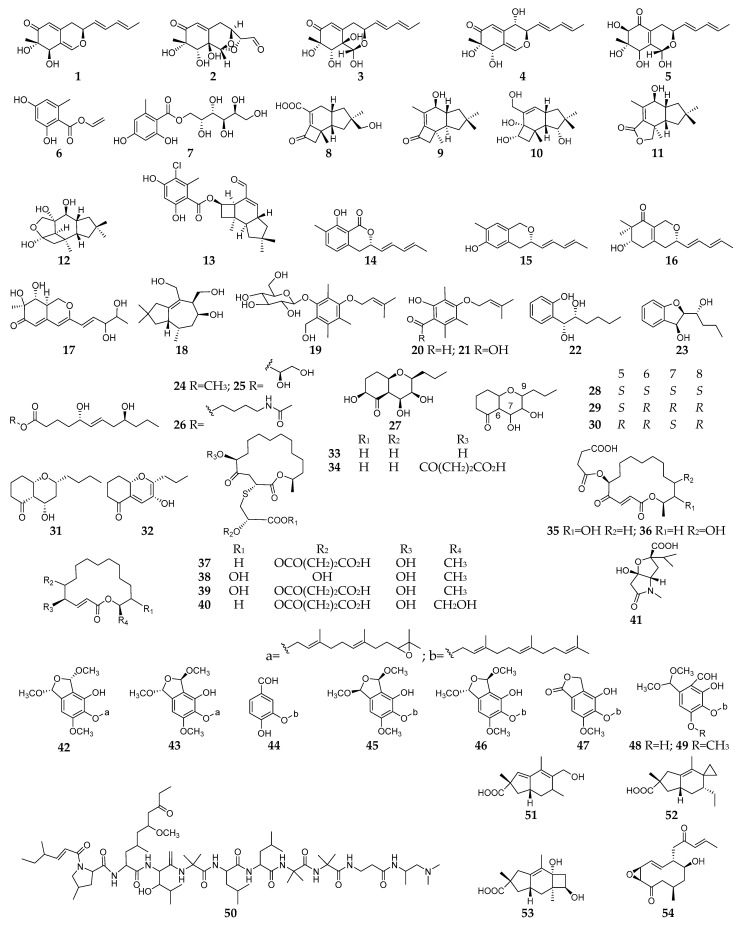
New SMs (**1**–**54**) isolated from fungal–fungal co-cultures using PDB.

**Figure 3 microorganisms-11-00464-f003:**
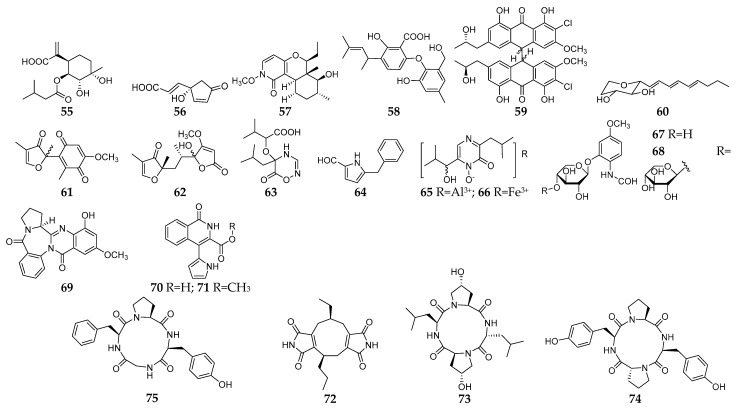
New SMs (**55**–**75**) isolated from fungal–fungal co-cultures using other liquid medium.

**Figure 4 microorganisms-11-00464-f004:**
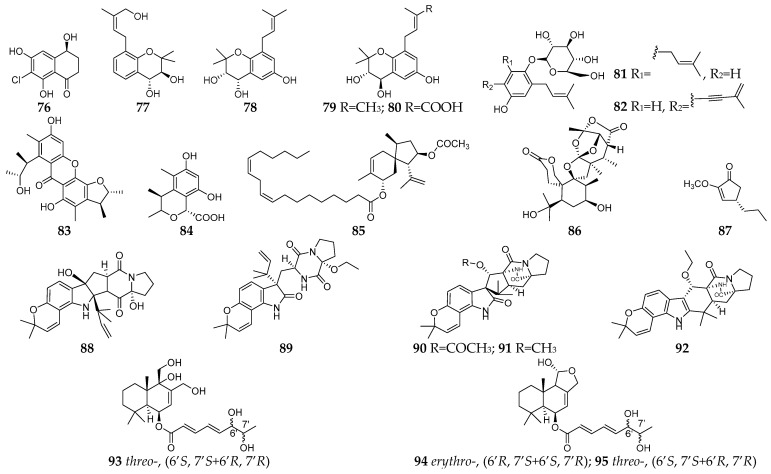
New SMs (**76**–**95**) isolated from fungal–fungal co-cultures using rice solid medium.

**Figure 5 microorganisms-11-00464-f005:**
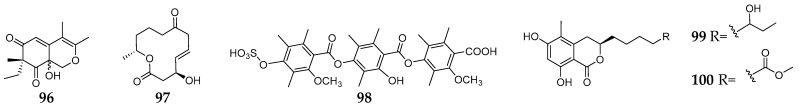
New SMs (**96**–**100**) were derived from fungal–fungal co-cultures using PDA.

**Figure 6 microorganisms-11-00464-f006:**
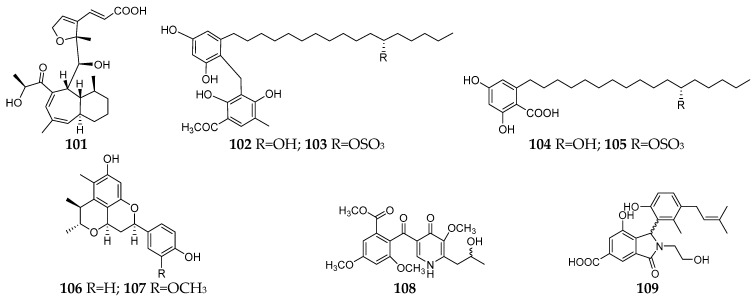
New SMs (**101**–**109**) isolated from fungal–fungal co-cultures using PDA.

**Figure 7 microorganisms-11-00464-f007:**
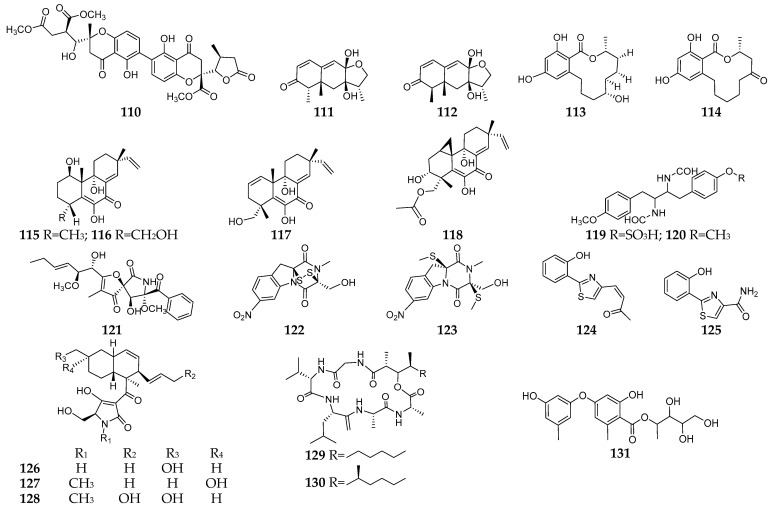
New SMs (**110**–**131**) isolated from fungal–bacterial co-culture using LSF.

**Figure 8 microorganisms-11-00464-f008:**
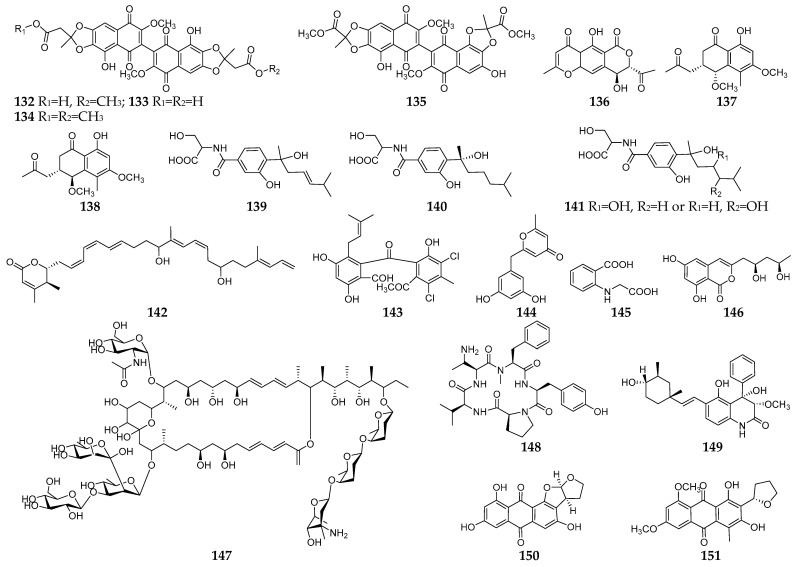
New SMs (**132**–**151**) isolated from fungal–bacterial co-culture using SSF.

**Figure 9 microorganisms-11-00464-f009:**
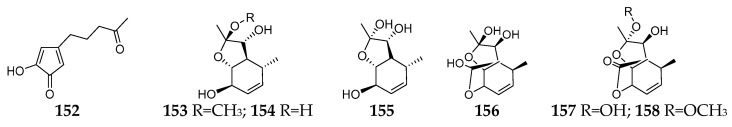
New SMs (**152**–**158**) isolated from fungal–host co-culture.

## Data Availability

Not applicable.

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
