# Peer review of "The Potential Use of Fungal Co-Culture Strategy for Discovery of New Secondary Metabolites"

_microorganisms, 2023, doi:10.3390/microorganisms11020464_

Round 1
Reviewer 1 Report
The manuscript "The Potential Use of Fungal Co-culture Strategy for Discovery of New Secondary Metabolites" is quite interesting for the area, because it is a compilation of substances produced by microorganisms co-culture.
The manuscript is very well written and well organized.
The authors carried out a survey of co-cultures between fungus-fungus, fungus-bacteria and fungus-host, highlighting the compounds produced through co-culture and their respective biological activities, in solid and liquid media.
In my opinion the English language is adequate. Only on line 211 page 7 there is the word in written twice
Author Response
Dear respected reviewer,
Thanks for your valuable comments on our manuscript (Microorganisms-2195808). According to your kind suggestions, the original work had been carefully revised, which were highlighted in red. Sincerely hope this improved work would be accepted for publication. Our point-to-point reply is as followings:
Q1. In my opinion the English language is adequate. Only on line 211 page 7 there is the word in written twice.
Our reply: Sorry for this error, which had been corrected.
Your kind assistance to improve our manuscript is greatly appreciated.
Huawei Zhang
Ph.D., professor of microbe natural products chemistry
School of Pharmaceutical Sciences
Zhejiang University of Technology
Hangzhou 310014
China
Reviewer 2 Report
In this manuscript the authors summarize the many fungal / fungal and bacterial / fungal interactions, that have been important for expression of specialized metabolites (SMs) that are otherwise often coded by silent gene clusters and not expressed. The use of different types of media / fermentation conditions for separate paragraphs is a good approach in my opinion, as otherwise it would be very long list of these interactions, and the production medium for SMs are often not taken into account in a serious way. There are some small problems that can easily be resolved:
1. The sentence: "These substances have played... (line 26-28) should be rewritten, as it seems that penicillin is a disease : " ... and other diseases, such as penicillin ..."
2. line 32 "expressed" rather than "express"
3. Line 40: "greater effect", greater that what?
4. Line 56: "respectively" (/what does respectively refer to?
5. Line 76: oxygen not oxyen
6. Line 108: When co-cultured not When co-culture
7. Line 127: Since you call the fungus Talaromyces pinophilus is used in line 102 (coorectly), why use Penicillium pinophilum in line 127?
8. Line 115: Be careful to use A. fumigatus, when you use Alternaria sp. in the same line, and Armillaria in line 82, rather write Aspergillus fumigatus
9. Please carefully go through all the other abbreviations for genera, to see if those abbreviations can be misunderstood
10. Line 207: B. eline, is that a correct species epithet?
11. Line 201: Maybe write out Penicillium crustosum
12. maybe write our Penicillium citrinum
13. Line 229: Is B. amyloliquefaciens Bacillus amyloliquefaciens?
14. Line 251: Maybe write Penicillium citrinum out
15 Line 240: Is S. peucetius Streptomyces peucetius?
16. Line 254: Fusarium palliudoroseum?
17 Line 280: What is "Wright actinomycete"??
Author Response
Dear respected reviewer,
Thanks for your valuable comments on our manuscript (Microorganisms-2195808). According to your kind suggestions, the original work had been carefully revised, which were highlighted in red. Sincerely hope this improved work would be accepted for publication. Our point-to-point reply is as followings:
Q1. The sentence: "These substances have played... (line 26-28) should be rewritten, as it seems that penicillin is a disease : " ... and other diseases, such as penicillin ..."
Our reply: Done as suggested. This confusing sentence had been rephrased.
Q2. line 32 "expressed" rather than "express"
Our reply: Done as suggested.
Q3. Line 40: "greater effect", greater that what?
Our reply: Microbial co-culture has a greater effect not only on microbe growth but also on metabolism than axenic culture.
Q4. Line 56: "respectively" (/what does respectively refer to?
Our reply: Based on fungal co-culture type, these new substances are respectively introduced herein and their detailed information is supplied in Tables S1-S3.
Q5. Line 76: oxygen not oxyen
Our reply: Done as suggested.
Q6. Line 108: When co-cultured not When co-culture
Our reply: Done as suggested.
Q7. Line 127: Since you call the fungus Talaromyces pinophilus is used in line 102 (correctly), why use Penicillium pinophilum in line 127?
Our reply: They belong to the same species but are different strains.
Q8. Line 115: Be careful to use A. fumigatus, when you use Alternaria sp. in the same line, and Armillaria in line 82, rather write Aspergillus fumigatus
Our reply: Done as suggested.
Q9. Please carefully go through all the other abbreviations for genera, to see if those abbreviations can be misunderstood
Our reply: Done as suggested.
Q10. Line 207: B. eline, is that a correct species epithet?
Our reply: B. eline is Beauveria felina.
Q11. Line 201: Maybe write out Penicillium crustosum
Our reply: Done as suggested.
Q12. maybe write our Penicillium citrinum
Our reply: Done as suggested.
Q13. Line 229: Is B. amyloliquefaciens Bacillus amyloliquefaciens?
Our reply: Yes and we have rewritten its full name.
Q14. Line 251: Maybe write Penicillium citrinum out
Our reply: Done as suggested.
Q15. Line 240: Is S. peucetius Streptomyces peucetius?
Our reply: Yes.
Q16. Line 254: Fusarium palliudoroseum?
Our reply: Yes.
Q17 Line 280: What is "Wright actinomycete"??
Our reply: It has been changed as “strain Actinomycete sp. WAC 2288”.
Your kind assistance to improve our manuscript is greatly appreciated.
Huawei Zhang
Ph.D., professor of microbe natural products chemistry
School of Pharmaceutical Sciences
Zhejiang University of Technology
Hangzhou 310014
China
Reviewer 3 Report
The review by S.Xu et al is devoted to the description of new secondary metabolites form during co-culture of fungi with other microorganisms.
It is really interesting and helpful. The MS completely corresponds to journal Microorganisms and is for sure of great interest to specialists dealing with secondary metabolites obtaining.
The authors conducted a qualitative literature search. The article can be accepted for publication in its present form. There is one typo: line 121 is missing a comma after glucose 10g/l.
what I lacked in this review: an analysis on what basis certain pairs were chosen for the cultivation of fungi. But I don't know if this question was part of the research area of ​​the author. otherwise, in my opinion, it is very useful.
Author Response
Dear respected reviewer,
Thanks for your valuable comments on our manuscript (Microorganisms-2195808). According to your kind suggestions, the original work had been carefully revised, which were highlighted in red. Sincerely hope this improved work would be accepted for publication. Our point-to-point reply is as followings:
Q1. line 121 is missing a comma after glucose 10g/l.
Our reply: Sorry for this error, which had been corrected.
Q2. An analysis on what basis certain pairs were chosen for the cultivation of fungi. But I don't know if this question was part of the research area of ​​the author. otherwise, in my opinion, it is very useful.
Our reply: Chemical investigation of fungal co-culture for new SM discovery is one of our research fields. Currently suitable paired strain for fungal co-culture is usually selected using metabolomics analysis, such as GNPS.
Your kind assistance to improve our manuscript is greatly appreciated.
Huawei Zhang
Ph.D., professor of microbe natural products chemistry
School of Pharmaceutical Sciences
Zhejiang University of Technology
Hangzhou 310014
China